# Expression of the Chemokine Receptor CCR7 in the Normal Adrenal Gland and Adrenal Tumors and Its Correlation with Clinical Outcome in Adrenocortical Carcinoma

**DOI:** 10.3390/cancers13225693

**Published:** 2021-11-14

**Authors:** Carmina Teresa Fuss, Katharina Other, Britta Heinze, Laura-Sophie Landwehr, Armin Wiegering, Charis Kalogirou, Stefanie Hahner, Martin Fassnacht

**Affiliations:** 1Division of Endocrinology and Diabetes, Department of Medicine I, University Hospital, University of Würzburg, 97080 Würzburg, Germany; Fuss_C@ukw.de (C.T.F.); Other_K@ukw.de (K.O.); Heinze_B@ukw.de (B.H.); Landwehr_L@ukw.de (L.-S.L.); Hahner_S@ukw.de (S.H.); 2Department of General, Visceral, Transplant, Vascular and Pediatric Surgery, University Hospital Würzburg, 97080 Würzburg, Germany; Wiegering_A@ukw.de; 3Theodor Boveri Institute, Biocenter, University of Würzburg, Am Hubland, 97074 Würzburg, Germany; 4Department of Urology, University Hospital Würzburg, 97080 Würzburg, Germany; Kalogirou_C@ukw.de

**Keywords:** CCR7, chemokine receptor, adrenocortical carcinoma, adrenal tumors

## Abstract

**Simple Summary:**

The chemokine receptor CCR7 plays an important role in immune function and lymphoid trafficking and has been shown to be expressed in several malignant tumors with evidence for a role in tumor cell progression and clinical outcome. Here, we investigated the expression of CCR7 in normal adrenal glands, adrenocortical adenomas, and adrenocortical carcinomas. CCR7 was expressed across all adrenal tissues. However, CCR7 protein or mRNA expression did not significantly influence patient survival in adrenocortical carcinoma, but it could play a role in adrenocortical homeostasis.

**Abstract:**

Background: The chemokine receptor CCR7 is crucial for an intact immune function, but its expression is also associated with clinical outcome in several malignancies. No data exist on the expression of CCR7 in adrenocortical tumors. Methods: CCR7 expression was investigated by qRT-PCR and immunohistochemistry in 4 normal adrenal glands, 59 adrenocortical adenomas, and 181 adrenocortical carcinoma (ACC) samples. Results: CCR7 is highly expressed in the outer adrenocortical zones and medulla. Aldosterone-producing adenomas showed lower CCR7 protein levels (H-score 1.3 ± 1.0) compared to non-functioning (2.4 ± 0.5) and cortisol-producing adenomas (2.3 ± 0.6), whereas protein expression was variable in ACC (1.8 ± 0.8). In ACC, CCR7 protein expression was significantly higher in lymph node metastases (2.5 ± 0.5) compared to primary tumors (1.8±0.8) or distant metastases (2.0 ± 0.4; *p* < 0.01). mRNA levels of CCR7 were not significantly different between ACCs, normal adrenals, and adrenocortical adenomas. In contrast to other tumor entities, neither CCR7 protein nor mRNA expression significantly impacted patients’ survival. Conclusion: We show that CCR7 is expressed on mRNA and protein level across normal adrenals, benign adrenocortical tumors, as well as ACCs. Given that CCR7 did not influence survival in ACC, it is probably not involved in tumor progression, but it could play a role in adrenocortical homeostasis.

## 1. Introduction

Adrenocortical carcinoma (ACC) is a rare cancer with an incidence of 0.7–2.0 per million per year [1,2]. Prognosis of ACC is generally poor but variable and treatment options are limited [3,4,5,6,7,8]. The main prognostic factors include tumor extension as described by ENSAT stage and the proliferation factor Ki67 [2,9,10]. 

Chemokines and their corresponding receptors are involved in inducing immune infiltration, cell migration, proliferation, and angiogenesis in various malignant tumors [11,12]. Recently, several reports have focused on the chemokine receptors CXCR4, as well as CXCR7 in ACC [13,14,15]. ACCs expressed both CXCR4 and CXCR7, but receptor expression did not impact patients’ survival [13]. 

The G-protein coupled chemokine receptor CCR7 is found on antigen-presenting dendritic cells, as well as B and T lymphocytes. CCL19 and CCL21 act as its only ligands and are mainly produced in lymph nodes [16]. CCR7 is essential in the development of lymph nodes, the functional organization of their follicles, and for guiding cells to lymphatic organs [17,18]. Several malignant tumors, such as lymphomas, breast, prostate, gastric, and colon cancer, show increased expression of CCR7 [19,20]. In patients with chronic lymphocytic leukemia, malignant cells migrate along a chemokine gradient of CCL19 and CCL21. Furthermore, CCR7 expression is correlated with clinical lymphadenopathy [19]. In a mouse model for metastatic breast cancer, CCR7-negative tumor cells metastasized to the lungs, whereas CCR7-positive transferred cells metastasized to lymph nodes and grew faster [21,22]. Similar results could be obtained for lymph node metastases in malignant melanoma [23]. CCR7 is also able to increase proliferation of cancer cells [24,25,26] as well as their stemness [27,28]. Of particular interest is that several CCR7 inhibitors are being tested in preclinical models [29]. 

To date, expression of CCR7 in adrenal tissues, and specifically its role in ACC, has not yet been assessed. In this study, we hypothesized that CCR7 is (1) expressed in adrenocortical tissues and, given its putative role in lymphatic dissemination of malignant tumors, that (2) CCR7 significantly impacts patient survival in ACC. 

## 2. Materials and Methods

### 2.1. Patients

We included biomaterial and clinical data from 120 patients with histologically confirmed ACC, APA (*n* = 20), CPA (*n* = 18), and NFA (*n* = 21). The following clinical and histopathological characteristics were assessed: sex, age at diagnosis, tumor size, resection status, Ki67 proliferation index, Weiss score, staging according to ENSAT classification [30], hormone secretion, and the presence of lymph node and distant metastases. Furthermore, response to mitotane and cytotoxic chemotherapy in patients with ACC was reported. However, prior to primary surgery, none of the patients had been treated medically. Patients had given written informed consent for tissue collection and analysis of clinical data. Histologically confirmed tumor-free normal adrenal glands were derived from surgery of patients with kidney cancer. They were used in anonymized fashion. The study was approved by the ethics committee of the University of Wuerzburg (No. 88/11). 

### 2.2. Gene Expression Analysis

Chemokine receptor mRNA expression levels were investigated by quantitative real-time polymerase chain reaction (qRT-PCR). RNA was isolated from fresh frozen tissue of ACCs (*n* = 9), normal adrenal glands (*n* = 4), aldosterone-producing adenomas (*n* = 11), cortisol-producing adenomas (*n* = 10), and non-functioning adenomas (*n* = 3), using the RNeasy Lipid Tissue Minikit (Qiagen, Hilden, Germany). Of the 33 tumor samples, 12 corresponded to samples investigated also by qRT-PCR (see below). Reverse transcription of RNA was performed using the QuantiTect Reverse Transcription Kit (Qiagen), as previously described [31]. We used the Taqman Gene Expression assays for CCR7 (Hs01013469_m1) from Applied Biosystems (Darmstadt, Germany). Endogenously expressed β-actin (Hs9999903_m1) was used for normalization. A quantity of 40 ng cDNA was used for each PCR reaction. Transcript levels were determined using the TaqMan Gene Expression Master Mix (Applied Biosystems), the CFX96 real-time thermocycler (Bio-rad, Hercules, CA, USA) and Bio-Rad CFX Manager 2.0 software. Cycling conditions were 95 °C for 3 min followed by 50 cycles of 95 °C for 30 s, 60 °C for 30 s, and 72 °C for 30 s. Using the ΔCT method, the gene expression levels were normalized to those of β-actin, as previously described [32].

Analysis of publicly available TCGA and GTEx data was performed using GEPIA2 [33,34,35]. For mRNA expression, analysis was based on “TCGA tumors versus (TCGA normal + GTEx normal)” and expression data was log2(TPM+1)-transformed. Pearson’s correlation of publicly available mRNA expression datasets was performed. 

### 2.3. Immunohistochemistry

Immunohistochemistry analysis was performed on 181 FFPE ACC specimens (120 primary tumors, 13 local recurrences, 29 lymph node metastases, 19 distant metastases), aldosterone-producing adenomas (*n* = 20), cortisol-producing adenomas (*n* = 18), non-functioning adenomas (*n* = 21), and 3 normal adrenal glands. The tissue sections were deparaffinized in xylene and rehydrated in ethanol (100, 90, 80, and 70% each concentration for 5 min). Immunohistochemical detection was performed using an indirect immunoperoxidase technique after high temperature antigen retrieval in 10 mM citric acid monohydrate buffer (pH 6.5) in a pressure cooker for 13 min. Blocking of unspecific protein-antibody interactions was performed with 20% human AB serum in PBS for 1 h at room temperature (RT). Primary CCR7 antibodies (Abcam, Camebridge, UK; ab32527) were used at a dilution of 1:3000 at RT for 1 h. Signal amplification was achieved by En-Vision System Labeled Polymer-HRP (Dako, Glostrup, Denmark) for 40 min and developed for 10 min with DAB Substrate Kit (Vector Laboratories, Burlingame, CA, USA) according to the manufacturer’s instructions. Mayer’s hematoxylin was used for the counterstaining of nuclei. Negative controls were carried out by treating the slides with N-Universal Negative Control Anti-Rabbit (Dako) instead of the primary antibody. Furthermore, tonsil served as the positive control and ovarian tissue as the negative control for CCR7 immunoreactivity. 

All slides were evaluated independently by two investigators blinded to patients’ clinical data. Staining intensity was evaluated with a grading score of 0, 1, 2, or 3, which corresponded to negative, weak, moderate, or strong staining intensity, respectively. The percentage of positive tumor cells was calculated for each specimen and scored 0 if 0% were positive, 0.1 if 1–9%, 0.5 if 10–49%, and 1 if ≥ 50%. A semi-quantitative H-score was then calculated by multiplying the staining intensity grading score with the proportion score as previously described [36]. Calculation of the H-score was performed separately for membrane and cytoplasmic staining. An H-score <2 was rated as low (weak staining), whereas an H-score ≥2 was rated as high (strong staining), for both membrane and cytoplasm. In cases of divergent results, slides were re-evaluated by all investigators, determining the final score by consensus.

### 2.4. Statistical Analysis

Statistical analysis was performed using SPSS Statistics Version 25 (IBM) and GraphPad Prism version 8.4.1 (GraphPad, La Jolla, CA, USA). Quantitative values were expressed as mean ± standard deviation or median and interquartile range as appropriate. Fisher’s exact or chi-square tests were used to analyze dichotomous variables, whereas continuous variables were investigated with a two-sided *t*-test or Mann-Whitney Test. Dunn’s multiple comparison test was used to determine differences in CCR7 mRNA expression and membrane staining between different adrenocortical tumors. Kaplan Meier survival analysis was used to investigate the correlation between CCR7 expression and prognosis. Progression-free survival (PFS) was defined as the time from the date of first surgery to the first radiological evidence of disease progression or death from ACC. Overall survival (OS) was defined as the time from the date of first diagnosis to the time of death or last follow-up. Disease-free survival (DFS) was defined as time from treatment to recurrent disease manifestation. Differences between survival curves were assessed by the log-rank (Mantel-Cox) test. Cox proportional hazard regression was used to analyze multiple potential influencing factors. Survival analysis of publicly available RNA-sequencing data from TCGA and GTEx was performed using GEPIA2 [33]. Hazard ratios were calculated based on the Cox proportional hazard model. Differences in survival were analyzed by log-rank test. *p*-values < 0.05 were considered statistically significant.

## 3. Results

### 3.1. Protein Expression of CCR7 in the Normal Adrenal, Adrenocortical Adenoma, and Adrenocortical Carcinoma 

First, we analyzed protein expression in normal adrenal glands and tumors of adrenocortical origin. Immunohistochemistry revealed a high and predominately membraneous expression of CCR7 in the outer adrenocortical zones as well as the adrenal medulla of normal adrenal glands (Figure 1A,B). Aldosterone-producing adenomas showed lower staining intensities for CCR7 compared to non-functioning and cortisol-producing adenomas (Figure 1G–M). In ACC, protein expression of CCR7 was highly variable (Figure 1C–F). On average, H-scores for membrane staining for CCR7 was significantly lower in ACCs compared to non-functioning adenomas (*p* < 0.01), whereas no significant differences could be detected between ACCs, aldosterone-producing, and cortisol-producing adenomas (Figure 1M). Looking not only at primary tumors but also recurrences, lymph node and distant metastases of ACCs, the H-score for CCR7 was significantly higher in lymph node metastases compared to primary tumors, recurrences, and distant metastases (all *p* < 0.01, Figure 2).

### 3.2. mRNA Expression of CCR7 in the Normal Adrenal, Adrenocortical Tumors, and Adrenocortical Carcinoma

Analysis of CCR7 mRNA levels in different adrenocortical tissues is shown in Figure 3A. Even though no significant differences could be detected between normal adrenal glands and adrenocortical tumors, normal adrenals and non-functioning adenomas showed the highest relative mRNA expression. Interestingly, ACCs also showed the lowest relative mRNA expression amongst different adrenocortical tissues (n.s., Figure 3A). We could confirm this expression pattern by investigating CCR7 expression in publicly available RNA sequencing datasets from TCGA [34] and GTEx [35], where ACC again had lower expression of CCR7, as well as its ligands CCL19 and CCL21 compared to normal adrenal glands (Figure 3B). Furthermore, CCR7 mRNA expression significantly positively correlated with CCL19 mRNA in ACCs and normal adrenals, whereas CCR7 was only significantly correlated with CCL21 mRNA expression in ACC but not normal adrenal tissue (Figure 3C,D). 

### 3.3. mRNA Expression of CCR7, CCL19 and CCL21 in Immune Cells of ACC and Normal Adrenal Glands

We subsequently analyzed publicly available data using GEPIA2021 and performed a sub-expression analysis with CIBERSORT deconvolution to look at CCR7, CCL19, and CCL21 mRNA in CD4+, CD8+, and regulatory T cells, as well as M1 and M2 macrophages in ACC and normal adrenal glands (Figure 4). CCR7, CCL19, and CCL21 mRNA levels differed depending on immune cell subtype in ACC and normal adrenals. However, in contrast to CCL19 and CCL21, only the expression pattern of CCR7 mRNA differed between ACC and normal adrenals. CCR7 was significantly lower in CD4+ and significantly higher in regulatory T cells and M1 macrophages of ACC compared to non-transformed adrenal tissue.

### 3.4. Predictive Role of CCR7 Staining Intensity and mRNA Expression on Clinical Outcome in Adrenocortical Carcinoma

Next, we correlated the CCR7 protein expression with various clinical characteristics of the tumors (Table 1). However, we could not detect any significant differences between patients with high and low H-score for CCR7 regarding ENSAT stage, resection status, hormone secretion, tumor size, Ki67 index, Weiss-Score, or presence of lymph node or distant metastases. Furthermore, neither progression-free survival (CCR7 low vs. high: 32 ± 37 vs. 25 ± 39 months, *p* = 0.318), nor overall survival (CCR7 low vs. high: 65 ± 48 vs. 61 ± 59 months, *p* = 0.632) were significantly different between patients with high or low CCR7 expression (Table 1, Figure 5). Multivariate Cox-regression analysis for age, Weiss-score, Ki67 index, CCR7 expression, ENSAT stage, and resection status only revealed Ki67 index and ENSAT stage, but not CCR7 to be significant prognostic factors for overall and progression-free survival (data not shown). To check whether CCR7 mRNA expression might influence patient survival in ACC, we re-analyzed the selected publicly available datasets. Consistent with our survival analysis regarding staining intensity, and therefore protein expression in ACC, we could not detect significant differences in disease-free and overall survival between patients with low and high CCR7 mRNA expression (Figure 6). Similarly, we could not find any significant differences in response to mitotane or cytotoxic chemotherapy between patients with high and low CCR7 expression. However, patients receiving treatment with mitotane (*n* = 82) showed a tendency towards longer overall survival in the case of low CCR7 expression compared to high CCR7 expression (81 months vs. 47 months, *p* = 0.06). 

## 4. Discussion

To date, ACC represents a rare cancer entity with limited prognosis and treatment options. Chemokine receptors have emerged as potential prognostic markers and interesting treatment targets in various tumor types. We, therefore, set out to investigate whether the chemokine receptor CCR7 plays a role in the adrenal gland, adrenal adenomas, and especially ACC.

The high expression of CCR7 in the normal adrenal gland both for mRNA as well as protein level suggests a potentially important physiological role. Interestingly, the receptor was localized mostly at the outer adrenocortical zone and in the adrenal medulla, whereas expression faded towards the zona reticularis of the adrenal cortex. Given the physiological role of CCR7 regarding directed cell migration and immune-cell trafficking, one can assume that CCR7, like CXCR4, may be involved in the functional zonation and differentiation of the adrenal cortex [13,16,37]. 

Looking at different adrenocortical adenomas, non-functioning and cortisol-producing adenomas showed the highest CCR7 expression both for mRNA and protein level in contrast to aldosterone-producing adenomas. This pattern seems somewhat surprising, because, presumably, aldosterone-producing cells in the zona glomerulosa of normal adrenal glands highly express CCR7. Therefore, CCR7 is most likely not involved or dependent on hormone production of different adrenal adenomas. Furthermore, in our analysis, CCR7 expression in ACC did not significantly differ when comparing glucocorticoid-secreting ACCs to tumors without glucocorticoid secretion. Conclusive evidence regarding the role of glucocorticoids in the regulation of CCR7 is still lacking. For example, glucocorticoids seem to downregulate CCR7 expression in dendritic cells, whereas administration of hydrocortisone in a randomized controlled trial did not lead to significant changes in CCR7 expression in CD4+ and CD8+ T cells [38,39]. However, further functional investigations are needed to elucidate the physiological role of CCR7 in the adrenal cortex, especially considering the microenvironment of the adrenal gland. In this context the variable mRNA expression of CCR7, CCL19, and CCL21 among the different immune cells is interesting. However, the high standard deviation prevents us from drawing a firm conclusion. 

In ACC, however, we observed lower mRNA levels of CCR7 compared to normal adrenals and adrenocortical adenomas and could confirm this finding in an independent cohort using publicly available datasets. In contrast, our immunohistochemical analyses revealed a variable immunoreactivity in ACC with an overall relatively high CCR7 membrane expression. At the moment, we can only speculate on this discrepancy. In theory, there could be an intrinsic factor that prolongs the stability of CCR7 protein in ACC and the high protein level could lead to a diminished transcription of CCR7. However, the abundant protein expression in ACC is consistent with previous data on other malignant diseases, such as breast, pancreatic, and gastric cancer [40,41,42,43,44]. Furthermore, lymph node metastases from ACCs had significantly higher CCR7 H-scores compared to primary tumors, recurrences, or distant metastases, suggesting a possible role in lymphatic spreading of tumor cells. This hypothesis would be consistent with studies in mouse models for esophageal and breast cancer. In both cases, CCR7-positive cell lines showed earlier lymphatic tumor spreading compared to CCR7-negative cells [21,45]. In contrast to several studies focusing on immunohistochemical expression of CCR7 in malignant tumors [40,41,42,43], we were not able to demonstrate significant differences in the rate of lymph node metastasis between patients with low and high CCR7 protein. In keeping with this, neither progression-free nor overall survival seemed to be influenced by CCR7, but rather by known factors such as ENSAT stage and Ki67 index, in our analysis. Again, these results could be replicated in an independent cohort using available mRNA results. 

Our study has several strengths and some drawbacks. The study is limited by its retrospective nature. Nonetheless, it is hypothesis-driven, and patients’ clinical data is very well annotated. We analyzed CCR7 expression in a large cohort of different normal, benign, and malignant adrenal tissues. However, the number of lymph node metastases, local recurrences and distant metastases was low compared to the number of primary tumors. To gain further insight into the physiological role of CCR7 in the adrenal gland, functional experiments, as well as consideration of the distribution of CCL19 and CCL21 in adrenal tissues, are needed, but this was beyond the scope of the present study. 

## 5. Conclusions

In summary, we show that CCR7 is expressed in mRNA and protein level across normal adrenal glands, different benign adrenocortical tumors, as well as ACCs. Given that CCR7 did not significantly influence survival of patients with ACC and its expression in other adrenal tissues, it is probable that CCR7 plays a role in adrenocortical homeostasis rather than tumor invasion and progression in ACC.

## Figures and Tables

**Figure 1 cancers-13-05693-f001:**
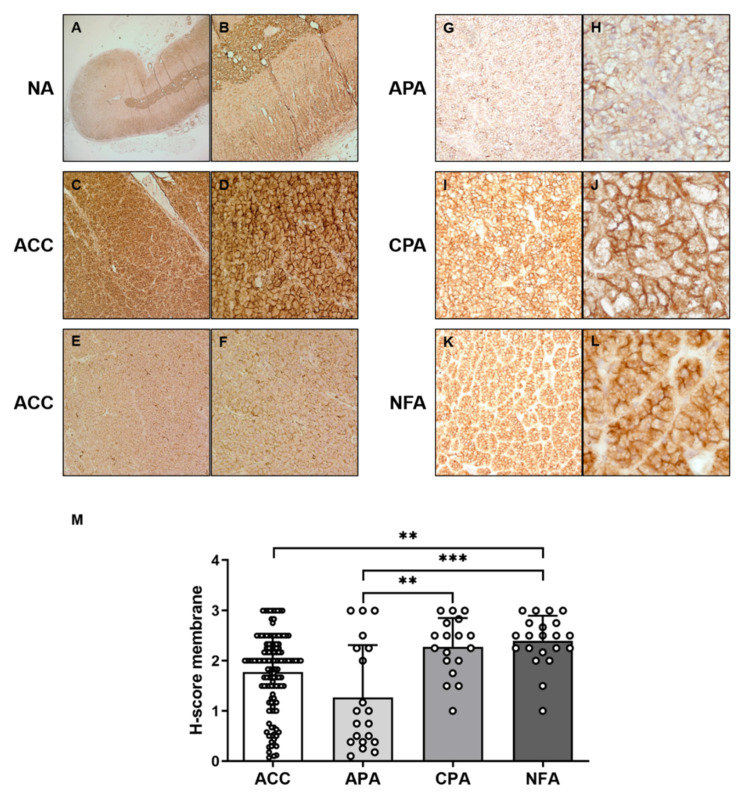
Representative immunohistochemical staining for CCR7 in the normal adrenal gland (NA). (**A**) 2.5× +, (**B**) 10×, adrenocortical carcinoma, (**C**) 10× +, (**D**) 40×, (**E**) 10× +, (**F**) 40×), aldosterone-producing adenoma (APA), (**G**) 10× +, (**H**) 40×), cortisol-producing adenoma (CPA), (**I**) 10× +, (**J**) 40×, non-functioning adenoma (**K**) 10× +, (**L**) 40×), (**M**) Comparison of membrane staining for CCR7 using H-score (mean ± SD) in ACC (*n* = 120), APA (*n* = 20), CPA (*n* = 18), and NFA (*n* = 21). ** *p* < 0.01, *** *p* < 0.001 (Dunn’s multiple comparisons test).

**Figure 2 cancers-13-05693-f002:**
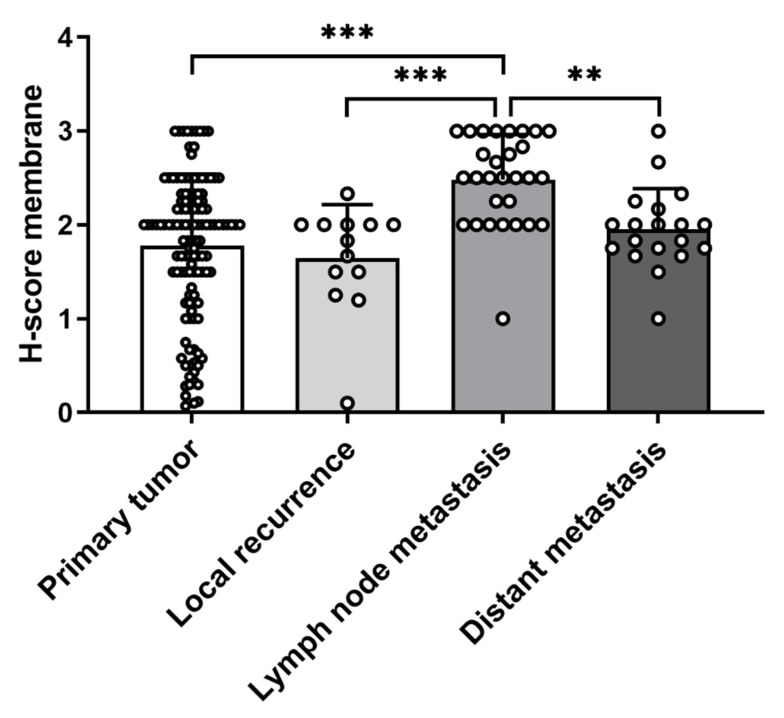
Comparison of membrane staining for CCR7 in adrenocortical carcinoma (primary tumors, *n* = 120), recurrences (*n* = 13), lymph node metastases (*n* = 29), and distant metastases (*n* = 19) using H-score (mean ± SD). ** *p* < 0.01, *** *p* < 0.001 (Dunn’s multiple comparisons test).

**Figure 3 cancers-13-05693-f003:**
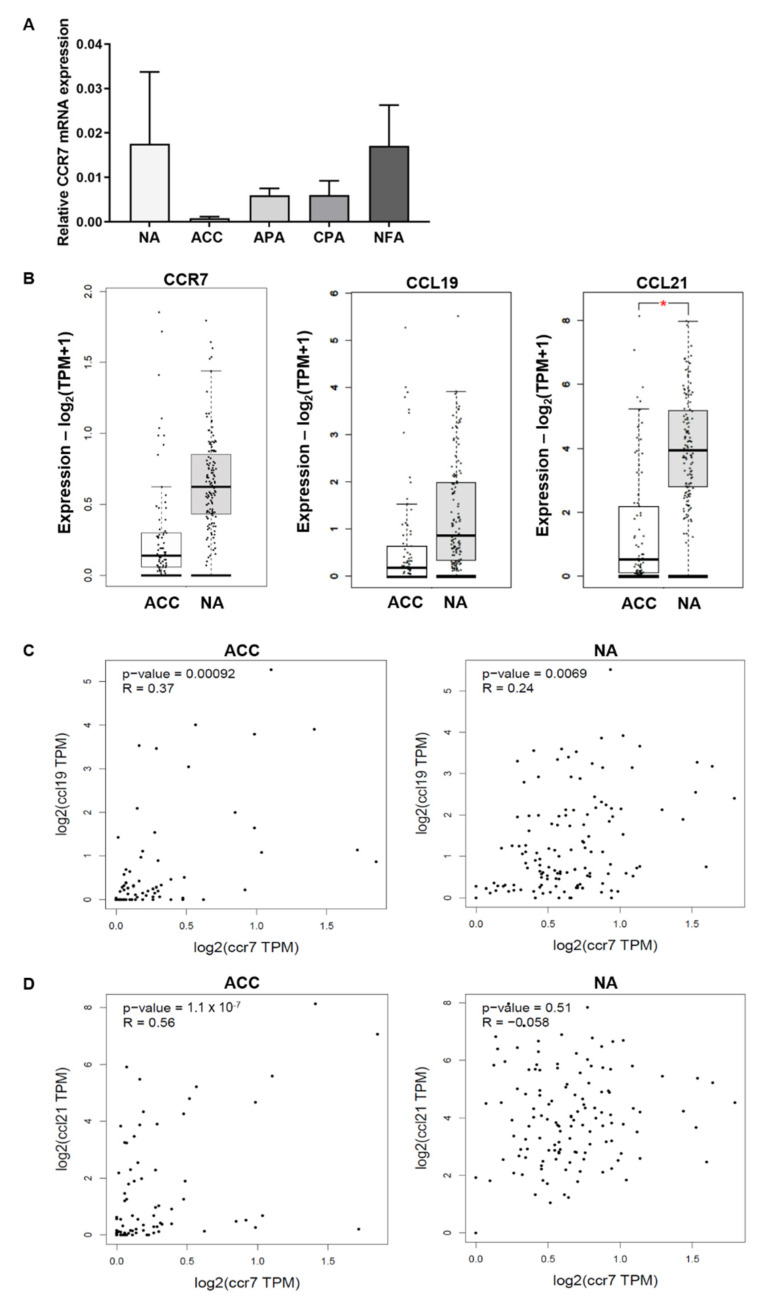
(**A**) Relative mRNA expression of CCR7 in normal adrenal (NA, *n* = 4), adrenocortical carcinoma (ACC, *n* = 9), aldosterone-producing adenoma (APA, *n* = 11), cortisol-producing adenoma (CPA, *n* = 10), and non-functioning adenoma (NFA, *n* = 3). Data is displayed as mean ± SEM. No significant differences were detected (Dunn’s multiple comparisons test). (**B**) mRNA expression data of CCR7 and its ligands CCL19 and CCL21 obtained from publicly available data sets using GEPIA2 (www.gepia2.cancer-pku.cn/; accessed 22 August 2021). Shown are adrenocortical carcinomas (ACC, *n* = 77) and normal adrenals (NA, *n* = 128). Significant differences were detected for CCL21 (one-way ANOVA). (**C**,**D**) Correlations of mRNA expression data for CCR7 and CCL19/21 for ACC and NA, respectively (GEPIA2, www.gepia2.cancer-pku.cn/, accessed 22 August 2021, Person’s correlation analysis).

**Figure 4 cancers-13-05693-f004:**
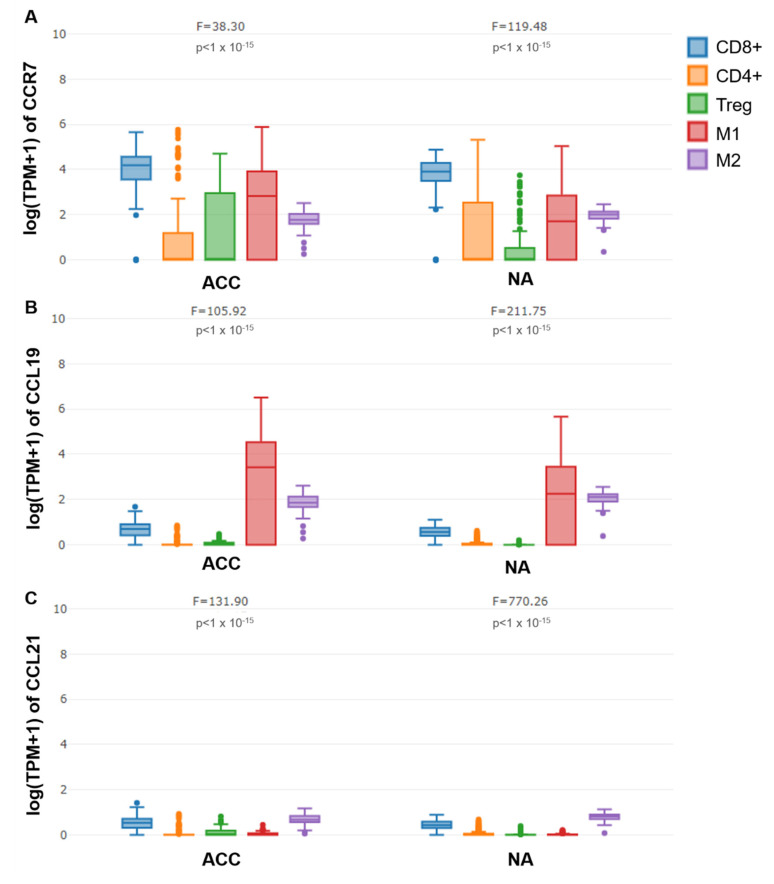
Comparison of CCR7 (**A**), CCL19 (**B**), and CCL21 (**C**) mRNA expression between CD8+, CD4+, and regulatory T cells, as well as M1 and M2 macrophages using publicly available datasets for adrenocortical carcinoma (ACC) and normal adrenal glands (NA). (GEPIA2021, sub-expression analysis, http://gepia2021.cancer-pku.cn/, accessed 22 August 2021; CIBERSORT deconvolution tool, one-way ANOVA).

**Figure 5 cancers-13-05693-f005:**
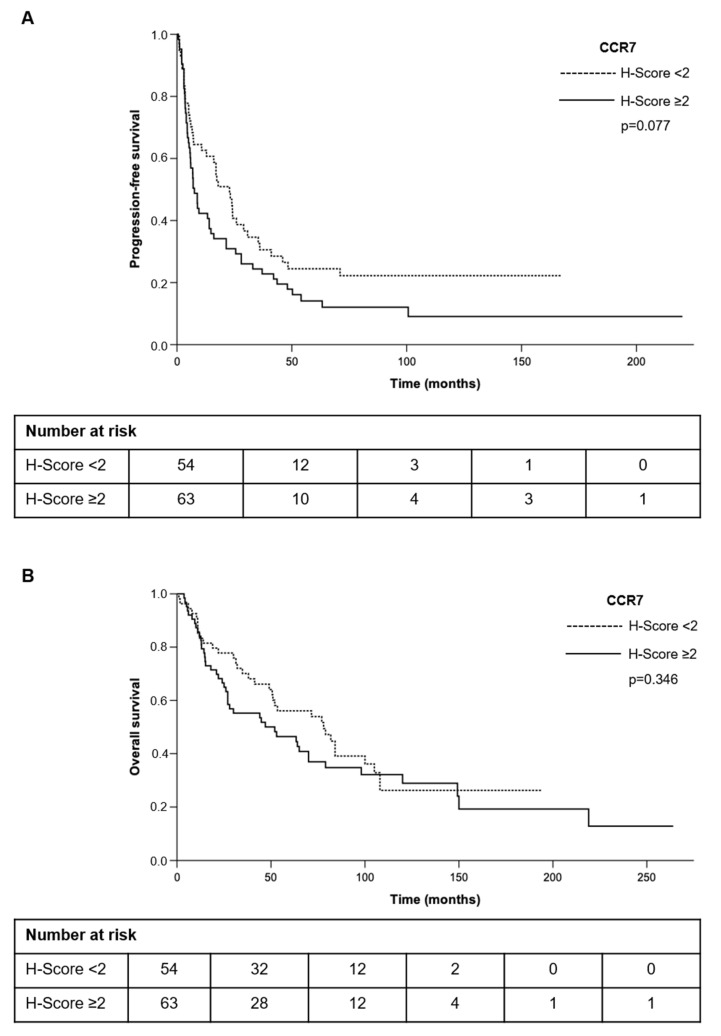
Progression-free survival (**A**) and overall survival (**B**) of patients with adrenocortical carcinoma influenced by CCR7 H-Score for membrane staining.

**Figure 6 cancers-13-05693-f006:**
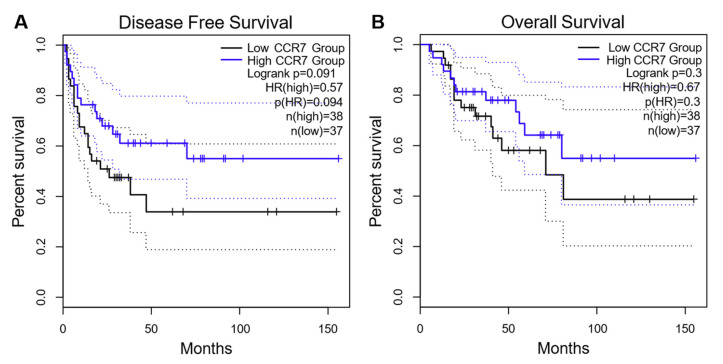
Disease free survival (**A**) and overall survival (**B**) of patients with adrenocortical carcinoma based on CCR7 mRNA expression using publicly available data sets. KM-plots were generated using GEPIA2 (www.gepia2.cancer-pku.cn/; accessed 22 August 2021). HR = Hazard Ratio.

**Table 1 cancers-13-05693-t001:** Characteristics of the ACC patients with analyzed primary tumors reported in this study.

	Total Cohort *n* = 120	H-Score < 2 *n* = 55	H-Score ≥ 2 *n* = 65	*p*-Value *
Age, years	50 ± 16	50 ± 16	50 ± 17	0.917
Female, *n* (%)	78 (65)	35 (64)	43 (66)	0.773
ENSAT stage, *n* (%)				0.477
I	5 (4)	2 (4)	3 (5)	
II	48 (42)	26 (48)	22 (35)	
III	35 (30)	13 (25)	22 (35)	
IV	27 (24)	12 (23)	15 (25)	
Resection status, *n* (%)				0.547
R0	59 (49)	27 (49)	32 (49)	
R1	6 (5)	4 (7)	2 (3)	
R2	23 (19)	12 (22)	11 (17)	
RX	9 (8)	6 (11)	3 (5)	
Unknown	23 (19)	6 (11)	17 (26)	
Autonomous hormone secretion, *n* (%)				0.646
GC	18 (15)	8 (14)	10 (15)	
GC + other	30 (25)	11 (20)	19 (29)	
MC	4 (3)	3 (6)	1 (2)	
A	8 (7)	3 (6)	5 (8)	
A + MC	1 (1)	0	1 (2)	
None	16 (13)	8 (14)	8 (12)	
No preoperative endocrine work-up	43 (36)	22 (40)	21 (32)	
Tumor size, cm	11.4 ± 4.8	11.3 ± 5.0	11.5 ± 4.6	0.881
Ki67 index, %	15 ± 12	14 ± 13	16 ± 11	0.340
Weiss-Score	5.7 ± 1.7	5.8 ± 1.6	5.6 ± 1.7	0.568
Lymph node metastasis, *n* (%)				
Overall	51 (43)	21 (38)	30 (46)	0.352
At primary diagnosis	19 (16)	6 (11)	13 (20)	0.091
At follow-up	36 (30)	17 (31)	19 (29)	0.722
Distant metastasis, *n* (%)	91 (76)	40 (73)	51 (78)	0.609
Disease progression, *n* (%)	95 (79)	40 (73)	55 (85)	0.068
Death overall, *n* (%)	78 (65)	34 (62)	44 (68)	0.431
Death due to ACC, *n* (%)	68 (57)	30 (55)	38 (59)	0.463
Median overall survival, months (±SD)	63 ± 53	65 ± 48	61 ± 59	0.632
Median progression-free survival, months (±SD)	28 ± 38	32 ± 37	25 ± 39	0.318

GC = glucocorticoids, MC = mineralocorticoids, A = androgens, ACC = adrenocortical carcinoma, * for comparison of H-Score < 2 vs. H-Score ≥ 2 (*t*-test and asymptotic chi-square test).

## Data Availability

The data presented in this study are available on request from the corresponding author. The data are not publicly available due to privacy issue due to the rarity of the disease.

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
