# Peer review of "Expression of the Chemokine Receptor CCR7 in the Normal Adrenal Gland and Adrenal Tumors and Its Correlation with Clinical Outcome in Adrenocortical Carcinoma"

_cancers, 2021, doi:10.3390/cancers13225693_

Round 1
Reviewer 1 Report
Authors examined CCR7 expression level in adrenocortical carcinoma, adenomas and normal adrenal glands by immunohistochemistry and qRT-PCR, comparing the findings with clinicopathological factors to clarify its possibility of being a potential prognostic marker. The results demonstrated that CCR7 immunoreactivity was higher in the foci of lymph node metastasis than in primary lesions although there was no significant difference of patient survival. As authors mentioned in introduction and discussion, CCR7 axis play roles in both aspects of tumor immune modulation and lymphatic spread.
The study contained a large number ACC cohorts with detailed clinical data. The results were novel and interesting but several points or questions were addressed here.
- CCR7 immunoreactivity was lowest in APAs and relatively higher in CPAs and NFAs rather than ACC. And CCR7 mRNA was higher in normal adrenal cortex than ACC. How could the authors explain the results above? And whether a possible association with in situ glucocorticoid excess should be considered or not. The relevant physiological microenvironment status of adrenal cortex should be also argued in order to explain the results above although the authors described that was beyond the aim of this study.
- Any findings or previous studies investigating the ligands (CCL19, CCL21) distribution in adrenocortical disorders would be requested, if possible.
- Are there any difference of CCR7 level regarding the responsiveness to adjuvant mitotane or chemotherapy?
- A matched analysis between primary- and LN-metastatic-lesions in the same cases would be requested if possible.
Author Response
We thank the reviewers for their careful review of our manuscript and for giving us the opportunity to improve our work. All changes made to the manuscript are highlighted in yellow.
The reviewers’ comments have been thoroughly addressed and we here provide our response in a point-by-point manner.
Reviewer 1:
Authors examined CCR7 expression level in adrenocortical carcinoma, adenomas and normal adrenal glands by immunohistochemistry and qRT-PCR, comparing the findings with clinicopathological factors to clarify its possibility of being a potential prognostic marker. The results demonstrated that CCR7 immunoreactivity was higher in the foci of lymph node metastasis than in primary lesions although there was no significant difference of patient survival. As authors mentioned in introduction and discussion, CCR7 axis play roles in both aspects of tumor immune modulation and lymphatic spread.
The study contained a large number ACC cohorts with detailed clinical data. The results were novel and interesting but several points or questions were addressed here.
RESPONSE: We appreciate the overall very positive judgment.
- CCR7 immunoreactivity was lowest in APAs and relatively higher in CPAs and NFAs rather than ACC. And CCR7 mRNA was higher in normal adrenal cortex than ACC. How could the authors explain the results above? And whether a possible association with in situ glucocorticoid excess should be considered or not. The relevant physiological microenvironment status of adrenal cortex should be also argued in order to explain the results above although the authors described that was beyond the aim of this study.
RESPONSE: We agree with the reviewer that these are interesting issues. However, we have to acknowledge that we cannot completely explain the described differences in CCR7 immunoreactivity and mRNA expression between the different tissues. However, it seems important to us to present all data as they are and not hide some of the results due to unexplained findings. Regarding the relevance of glucocorticoid excess, current evidence suggests that glucocorticoids downregulate CCR7 expression for example in dendritic cells (https://doi.org/10.1189/jlb.1110615), whereas in a randomized controlled trial the administration of hydrocortisone did not lead to significant changes of CCR7 expression on CD4+ and CD8+ T cells (https://doi.org/10.1152/ajpendo.00678.2013).
In line with your comment, we analyzed CCR7 protein abundance based on glucocorticoid secretion by the tumors and also could not find significant differences in H-Scores (median membranous H-Score glucocorticoid-secreting ACCs 2.0 vs. no glucocorticoid-secreting ACCs 1.8, p=0.399).
We have added the following section to the Discussion of the manuscript:
“Furthermore, in our analysis, CCR7 expression in ACC did not significantly differ when comparing glucocorticoid-secreting ACCs to tumors without glucocorticoid secretion. Conclusive evidence regarding the role of glucocorticoids in the regulation of CCR7 is still lacking. For example, glucocorticoids seem to downregulate CCR7 expression in dendritic cells, whereas administration of hydrocortisone in a randomized controlled trial did not lead to significant changes in CCR7 expression on CD4+ and CD8+ T cells [38, 39]. However, further functional investigations are needed to elucidate the physiological role of CCR7 in the adrenal cortex, especially taking into account the microenvironment of the adrenal gland.”
- Any findings or previous studies investigating the ligands (CCL19, CCL21) distribution in adrenocortical disorders would be requested, if possible.
RESPONSE: Thank you for this valuable comment. We performed a literature search and did not find any matching results. However, we analyzed publicly available datasets regarding CCL19 and CCL21 expression and added the results to the manuscript (Figure 3 B-D).
- Are there any difference of CCR7 level regarding the responsiveness to adjuvant mitotane or chemotherapy?
RESPONSE: To address the question of the Reviewer, we analyzed the response to mitotane or chemotherapy in our cohort and did not find significant differences in overall survival between patients with high and low CCR7 expression. However, patients receiving treatment with mitotane (n=82) showed a tendency towards longer overall survival in case of low CCR7 expression compared to high CCR7 (81 months vs 47 months, p=0.06). We added now a short sentence in the Methods:
"Furthermore, response to mitotane and cytotoxic chemotherapy in patients with ACC was reported."
and two sentences in the Results:
"Similarly, we could not find any significant differences in response to mitotane or cytotoxic chemotherapy between patients with high and low CCR7 expression. However, patients receiving treatment with mitotane (n=82) showed a tendency towards longer overall survival in case of low CCR7 expression compared to high CCR7 (81 months vs 47 months, p=0.06)."
- A matched analysis between primary- and LN-metastatic-lesions in the same cases would be requested if possible.
RESPONSE: Unfortunately, we have only 4 sample pairs of primary tumors and lymph node metastases in our cohort. We listed membrane H-Scores of those pairs in the table below. In those cases, CCR7 expression is higher for lymph node metastases compared to primary tumors, which is in accordance to the unmatched group comparison.
|
Patient # |
1 |
2 |
3 |
4 |
|
H-Score primary tumor |
1.6 |
2.3 |
0 |
1.7 |
|
H-Score lymphatic metastasis |
2.8 |
2.3 |
2 |
2.5 |
Due to the low number of patients, we would suggest not to include this data in the manuscript.
Reviewer 2 Report
This is an interesting study on the expression of chemokine CCR7 receptor in adrenocortical tumors. The authors included a large number of tumors for immunohistochemistry, and also performed mRNA-expression studies.
Differences of CCR7 protein and mRNA expression were noted among different subsets of adrenocortical tumors, but CCR7 expression could not be shown to affect survival of ACC.
The authors have shown that protein expression of CCR7 is variable, but lower than in non-functioning and cortisol-producing adenomas. On the other hand, the mRNA expression in ACC was considerably lower than in adenomas. How do the authors explain the difference of mRNA and protein expression, i.e. mRNA expression is much lower than that of protein?
The CCR7 expression was lower in aldosterone-producing adenomas than in cortisol-producing adenomas.
It would be interesting to divide the group of ACC into cortisol-producing and hormonally inactive tumors and to compare their CCR7 expression.
Are there data on the regulation of CCR7 expression by hormones, e.g. by cortisol or aldosterone?
Author Response
We thank the reviewers for their careful review of our manuscript and for giving us the opportunity to improve our work. All changes made to the manuscript are highlighted in yellow.
The reviewers’ comments have been thoroughly addressed and we here provide our response in a point-by-point manner.
Reviewer 2:
This is an interesting study on the expression of chemokine CCR7 receptor in adrenocortical tumors. The authors included a large number of tumors for immunohistochemistry, and also performed mRNA-expression studies.
RESPONSE: We are thankful for this positive statement.
Differences of CCR7 protein and mRNA expression were noted among different subsets of adrenocortical tumors, but CCR7 expression could not be shown to affect survival of ACC.
The authors have shown that protein expression of CCR7 is variable, but lower than in non-functioning and cortisol-producing adenomas. On the other hand, the mRNA expression in ACC was considerably lower than in adenomas. How do the authors explain the difference of mRNA and protein expression, i.e. mRNA expression is much lower than that of protein?
RESPONSE: We fully agree with the reviewer that this is indeed an interesting, somewhat puzzling observation that we cannot really explain, but only speculate. We added now the following sentences to the Discussion: "Right now, we can only speculate on this discrepancy. In theory, there could be an intrinsic factor that prolongs the stability of CCR7 protein in ACC and the high protein level could lead to an diminished transcription of CCR7."
The CCR7 expression was lower in aldosterone-producing adenomas than in cortisol-producing adenomas.
It would be interesting to divide the group of ACC into cortisol-producing and hormonally inactive tumors and to compare their CCR7 expression.
RESPONSE: Thank you for your comment. We analyzed our dataset accordingly and compared CCR7 membrane staining intensity between glucocorticoid- and non-glucocorticoid-secreting tumors. However, we could not detect significant differences between the two groups (median H-Score 2.0 vs. 1.8, p=0.399). We added one sentence to the Discussion:
"Furthermore, in our analysis, CCR7 expression in ACC did not significantly differ when comparing glucocorticoid-secreting ACCs to tumors without glucocorticoid secretion."
Are there data on the regulation of CCR7 expression by hormones, e.g. by cortisol or aldosterone?
RESPONSE: There is indeed not much published data regarding hormonal regulation of CCR7 expression. However, we have added the following section to the manuscript:
“Evidence regarding the role of glucocorticoids in the regulation of CCR7 is still lacking. For example, glucocorticoids seem to downregulate CCR7 expression in den-dritic cells, whereas administration of hydrocortisone in a randomized controlled trial did not lead to significant changes in CCR7 expression on CD4+ and CD8+ T cells [38, 39]. However, further functional investigation is needed to elucidate the physiological role of CCR7 in the adrenal cortex, especially taking into account the microenvironment of the adrenal gland.”
Reviewer 3 Report
This manuscript by Carmina et. al addresses CCR7 expression and its impact on patient survival in adrenocortical tissues. As we all know, CCR7 promotes lymphocyte homing to lymph nodes but there is increasing evidence of the chemokine receptor playing a role in cancer cell dissemination and metastasis. Interestingly, the authors hint at this and support the idea by showing CCR7 expression significantly higher in lymph node metastasis. There is also evidence of the chemokine receptor playing a role directly in cancer cell survival. This should be mentioned and appropriately cited in the intro, even if it is not in adrenocortical cancers.
The authors seek to look at the interplay by measuring both mRNA and protein expression of CCR7 in tissue and public databases.
While the authors show that CCR7 itself may not predict patient survival an indirect mechanism involving CCR7 axis can. For example, if CCL19 and CCL21 (CCR7 ligands that would presumably retain the CCR7 expressing cancer cells in the tumor) are present and higher in certain patients than this can also impact other immune cell migration into the tissue, including both good and bad players the antitumor immune response. Along those lines, I would recommend to look at: CCL21 and CCL19 expression as well as other immune cells trafficking (especially CD8 T cells, CD4 T cells, macrophages, regulatory T cells). I would also correlate the immune trafficking with expression of the ligands and CCR7. While the best way would be to look at all these cells and markers in your tissue, if not feasible, at least correlations and expression of markers from public databases is warranted.
The discussion sufficiently addressed all points.
Author Response
We thank the reviewers for their careful review of our manuscript and for giving us the opportunity to improve our work. All changes made to the manuscript are highlighted in yellow.
The reviewers’ comments have been thoroughly addressed and we here provide our response in a point-by-point manner.
Reviewer 3:
This manuscript by Carmina et. al addresses CCR7 expression and its impact on patient survival in adrenocortical tissues. As we all know, CCR7 promotes lymphocyte homing to lymph nodes but there is increasing evidence of the chemokine receptor playing a role in cancer cell dissemination and metastasis. Interestingly, the authors hint at this and support the idea by showing CCR7 expression significantly higher in lymph node metastasis. There is also evidence of the chemokine receptor playing a role directly in cancer cell survival. This should be mentioned and appropriately cited in the intro, even if it is not in adrenocortical cancers.
RESPONSE: Thank you for your comment. We have expanded the Introduction section of the manuscript accordingly: "CCR7 is also able to increase proliferation of cancer cells [24-26] as well as their stemness [27, 28]."
The authors seek to look at the interplay by measuring both mRNA and protein expression of CCR7 in tissue and public databases.
While the authors show that CCR7 itself may not predict patient survival an indirect mechanism involving CCR7 axis can. For example, if CCL19 and CCL21 (CCR7 ligands that would presumably retain the CCR7 expressing cancer cells in the tumor) are present and higher in certain patients than this can also impact other immune cell migration into the tissue, including both good and bad players the antitumor immune response. Along those lines, I would recommend to look at: CCL21 and CCL19 expression as well as other immune cells trafficking (especially CD8 T cells, CD4 T cells, macrophages, regulatory T cells). I would also correlate the immune trafficking with expression of the ligands and CCR7. While the best way would be to look at all these cells and markers in your tissue, if not feasible, at least correlations and expression of markers from public databases is warranted.
RESPONSE: Thanks a lot for your suggestions. We analyzed publicly available data regarding CCL19 and CCL21 expression in ACC and the normal adrenal, as well as CCR7/CCL19/CCL21 in immune cells in ACC and normal adrenals. We have therefore modified Figure 3 (Panel B-D) and added a completely new Figure 4 as well as an entire new paragraph to address the question of CCR7, CCL21 and CCL19 expression in different immune cells:
"3.3. mRNA expression of CCR7, CCL19 and CCL21 in immune cells of ACC and normal adrenal glands
We subsequently analyzed publicly available data using GEPIA2021 and performed a sub-expression analysis with CIBERSORT deconvolution to look at CCR7, CCL19 and CCL21 mRNA in CD4+, CD8+ and regulatory T cells, as well as M1 and M2 macrophages in ACC and normal adrenal glands (Figure 4). CCR7, CCL19 and CCL21 mRNA levels differed depending on immune cell subtype in ACC and normal adrenals. However, in contrast to CCL19 and CCL21, only the expression pattern of CCR7 mRNA differed between ACC and normal adrenals. CCR7 was significantly lower in CD4+ and significantly higher in regulatory T cells and M1 macrophages of ACC compared to non-transformed adrenal tissue."
However, due to the high standard deviation we are very cautious in the Discussion:
"In this context the variable mRNA expression of CCR7, CCL19 and CCL21 among the different immune cells is interesting. However, the high standard deviation prevents us from strong conclusion."
Reviewer 4 Report
The authors have analyzed the expression of the chemokine receptor CCR7 in normal adrenal gland and adrenal tumors at the RNA and protein levels on a retrospective cohort. This study raised some questions:
- I don't understand why the authors have decided to focus on such chemokine receptor in this cancer, as they could have picked up other receptors, which might explain why the results are mostly negative.
- It is not indicated how the selection of patients was done. Were the patients untreated or some of them were treated before surgery. For the follow-up of the patients, they certainly receive different types of treatments. So, the patients should be divided according to this criteria.
- What was the percentage of tumoral tissues in the samples used to extract RNA?
- What was the limit of detection set in real-time PCR in terms of Cp? It is mentioned that 50 cycles were performed, which is very high. What was the average of detection in Cp for CCR7?
- Do the tumors used for IHC and PCR correspond to the same patients?
- They authors should provide adequate controls of specificity of the anti-CCR7 antibody used with negative and positive controls, at least on cell lines expressing or not the receptor. It is surprising to detect a cytoplasmic expression of such receptor.
- A Pearson's Chi-square statistical test should be used to compare the different types of tumors.
Author Response
We thank the reviewers for their careful review of our manuscript and for giving us the opportunity to improve our work. All changes made to the manuscript are highlighted in yellow.
The reviewers’ comments have been thoroughly addressed and we here provide our response in a point-by-point manner.
Reviewer 4:
The authors have analyzed the expression of the chemokine receptor CCR7 in normal adrenal gland and adrenal tumors at the RNA and protein levels on a retrospective cohort. This study raised some questions:
- I don't understand why the authors have decided to focus on such chemokine receptor in this cancer, as they could have picked up other receptors, which might explain why the results are mostly negative.
RESPONSE: Based on our previous work on chemokine receptors in ACC and other adrenocortical tumors (CXCR4 and CXCR7; Bluemel et al. 2017, Chifu et al. 2020), we were intrigued by CCR7 as it has been described to influence tumor metastasis and lymphoid trafficking. Given that we also have a strong expertise regarding the immune cell environment in ACC (e.g. Landwehr L et al., J Immunother Cancer 2020) we conducted a hypothesis-driven study rather than following an unbiased approach.
- It is not indicated how the selection of patients was done. Were the patients untreated or some of them were treated before surgery. For the follow-up of the patients, they certainly receive different types of treatments. So, the patients should be divided according to this criteria.
RESPONSE: We checked our ACC biobank for available tumor material and included all patients with available FFPE samples and sufficient clinical data up to the time-point of the study. All patients were medically untreated at the time of the primary surgery. We added the following statement in the Methods:
"However, prior the primary surgery none of the patients have been treated medically."
Regarding any correlation of CCR7 expression and response to treatment, we would like to refer to our response to comment #3 of Reviewer 1.
- What was the percentage of tumoral tissues in the samples used to extract RNA?
RESPONSE: We did not investigate the tumor purity of the samples. However, it is well established that ACC has a very high tumor purity usually clearly > 80% (Zheng et al. Cancer Cell 2016).
- What was the limit of detection set in real-time PCR in terms of Cp? It is mentioned that 50 cycles were performed, which is very high. What was the average of detection in Cp for CCR7?
RESPONSE: The real-time PCR settings used correspond to our in-house standard protocols. However, we do agree that for detection of the CCR7 amplicon, fewer cycles would have been sufficient. The average of detection regarding Cp values across all tumor types was 37. In order to confirm the mRNA expression pattern between normal adrenal glands and ACCs, we decided to analyze publicly available datasets that showed similar results (as shown in Figure 3).
- Do the tumors used for IHC and PCR correspond to the same patients?
RESPONSE: No, not all tumors correspond to the same patients. In detail, 12 of 33 tumoral samples analyzed via qPCR were also investigated by IHC. This information is now given in the Methods:
"12 of these 33 tumor samples corresponded to samples investigated also by qRT-PCR."
- They authors should provide adequate controls of specificity of the anti-CCR7 antibody used with negative and positive controls, at least on cell lines expressing or not the receptor. It is surprising to detect a cytoplasmic expression of such receptor.
RESPONSE: We have performed antibody validation prior to the study. Results of our validation are shown below. Regarding cytoplasmatic detection, one can assume that even though CCR7 needs to be on the cell membrane in order to function, it is subject to regular degradation and recycling and therefore also present in the cytoplasm.
We added the following sentence in the Methods:
"Furthermore, tonsil served as positive control and ovarian tissue as negative control for CCR7 immunoreactivity."
Fig. 1: Immunohistochemical staining of CCR7 in the tonsil (positive control, A+B) and ovarian tissue (negative control, C+D).
Fig. 2: Side-by-side comparison of Western blot analysis and immunohistochemistry of healthy lung (A) and gastric (B) tissue.
- A Pearson's Chi-square statistical test should be used to compare the different types of tumors.
RESPONSE: Here we do not agree with the Reviewer. According our statistical adviser for comparison of non-parametric parameters the Dunn’s multiple comparison test is a very appropriate test.

Round 2
Reviewer 4 Report
The authors have answered to some of the points raised, but overall this does not modify consequently the manuscript. Here are my concerns:
- The justification of looking at CCR7 remains poor and cannot be justified by previous studies on CXCR4 and CXCR7.
- The assessment of the percentage of tumor tissue remains essential, especially when using RNA from whole tumors
- An average Cp value of 37 is very high and it is difficult to consider a significant expression of CCR7 in such conditions.
- The cytoplasmic staining by CCR7 antibody is intriguing and without a clear control with cell lines expressing or not CCR7, the specificity of the antibody remains questionable.